# Improving Lignocellulosic and Non-Lignocellulosic Biomass Characteristics through Torrefaction Process

Maja Ivanovski [1,2], Danijela Urbancl [1], Aleksandra Petrovič [1], Janja Stergar [1], Darko Goričanec [1] and Marjana Simonič [1,*]

[1] Faculty of Chemistry and Chemical Engineering, University of Maribor, Smetanova 17, 2000 Maribor, Slovenia
[2] Milan Vidmar Electric Power Research Institute, Department for Environment, Hajdrihova 2, 1000 Ljubljana, Slovenia
* Correspondence: marjana.simonic@um.si

**Abstract:** In this study, three locally available biomasses, namely miscanthus, hops, sewage sludge, and additionally, their mixtures, were subjected to the torrefaction process to improve their fuel properties. The torrefaction process was conducted at 250–350 °C and 10–60 min in a nitrogen ($N_2$) environment. The torrefaction temperature and time were studied to evaluate the selected biomass materials; furthermore, heating values, mass and energy yields, enhancement factors, torrefaction severity indexes (TSI), and energy-mass co-benefit indexes (EMCI) were calculated. In addition, thermogravimetric (TGA) and Fourier transform infrared analyses (FTIR) were performed to characterize raw and torrefied biomass under the most stringent conditions (350 °C and 60 min). The results showed that with increasing torrefaction temperature and duration, mass and energy yields decreased, and heating values (HHVs) increased for all studied biomasses. The results of the TSI and EMCI indexes showed that the optimum torrefaction conditions were as follows: 260 °C and 10 min for pure miscanthus and hops, whilst this could not be confirmed for the sewage sludge. Furthermore, the combination of sewage sludge and the above-mentioned types of lignocellulosic biomass exhibited better fuel properties than sewage sludge alone.

**Keywords:** lignocellulosic biomass; sewage sludge; torrefaction; thermogravimetry; TSI



## 1. Introduction

Today's world is still mostly relying on fossil fuels to satisfy its energy demands. The last two years have been marked globally by the COVID-19 pandemic, the recovery of economies after it, increased demand for energy supplies (mainly natural gas and coal), and the war in Ukraine. These events led to highly volatile, complicated, and risky energy markets. Hence, due to lower gas inflow to the European Union (EU) and increased demand for coal from China, as a result of higher electricity consumption, the prices for natural gas and coal began to rise [1]. On this account and on account of satisfying the environmental requirements set out in the European Green Deal [2], studying the potential and characteristics of renewable energy sources (RES), such as wind, solar, geothermal, marine energy, and hydropower, became necessary [3,4].

As one of the smallest EU countries, the Republic of Slovenia, has set an overarching national goal of achieving at least a 27% share of RES (mostly wind, solar, and hydro energy) in gross final energy consumption by 2030 [5]. In accordance with the National Energy and Climate Plan [6] prepared by the Ministry of Infrastructure in 2019, the sectoral target shares of RES by 2030 are as follows: 41.4% for the heating and cooling sector, 43.3% for the electricity sector, and 20.8% for the transport sector, of which at least 11% is the share of biofuels. Additionally, in the field of decarbonization, Slovenia has designated a mission of reducing total greenhouse gas emissions by 36% by 2030 [6]. Recently, biomass energy has been recognized as a promising sustainable and carbon-neutral source in the

country that can help to reduce the impact of social and economic outbursts [7,8] and also produce biofuels [9,10].

Due to high moisture content and low bulk density, as well as low calorific value, raw biomass is characterized by low-quality final products [11,12]. Pretreatment processes of raw biomass may also be required, which is connected with economic factors [13,14].

Numerous pretreatment processes have been investigated to date; most of them are related to thermochemical conversion processes such as pyrolysis [15], combustion [16], gasification [17], or liquefaction [18]. Raw lignocellulosic and non-lignocellulosic biomass materials, such as coffee grounds [19], miscanthus [20], rice husk [21], municipal solid waste [22], or sewage sludge [23] have already been successfully applied to the aforementioned thermochemical processes. In particular, the main components of lignocellulosic biomass are cellulose, hemicellulose, and lignin, followed by various proportions of minerals, moisture, and proteins [24]. It can be mostly used for green fuel production [25], as it is more or less derived from natural jungles, farms, and agricultural wastes [26,27]. Furthermore, it is characterized as the most abundant biomass on the planet, with an annual production of over 100 billion tonnes [28]. In Slovenia, this proportion is 8,419,970 m$^3$ per year or 7.10 m$^3$ per hectare [29], as Slovenia is fourth among the woodiest countries in the EU [29]. Even though forests cover approximately 58% of the country's area, municipal solid waste, industrial waste, or sewage sludge could replace lignocellulosic biomass in thermochemical processes [30,31] due to environmental friendliness and cost-effectiveness [32,33]. Sewage sludge (SS) co-combustion has been studied recently for energy utilization while improving organic compounds share [34,35]. Sewage sludge is an incidental product generated in biological wastewater treatment plants [36]. The content of dry matter is in the range of 60–70% during aerobic treatment [37,38]. Generally, it consists of various complex components due to its diverse sources, from garbage, work activity, and industrial processes [39]. Nowadays, the sludge is landfilled [40], gasified, or pyrolyzed [41]. Every year, up to 10 million tonnes of dry sewage sludge are produced worldwide [42], and the number is still increasing due to the growing urban population and urbanization [43]. Legislation related to sewage sludge management has been very strict to ensure safe sludge usage in industry, especially in the EU, since the implementation of the new Urban Waste Water Treatment Directive (91/271/CEE) [44], which has also brought an increase in the installation of the wastewater treatment plants.

Torrefaction is a promising biomass degradation step at increased temperature (200 to 300 °C) in anaerobic conditions and non-increased pressure. During the process, solid material with a high amount of carbon is produced [45]. The process has been recognized as one of the most feasible ways to upgrade the physical and chemical properties of raw lignocellulosic and non-lignocellulosic biomass materials [22,46] and, subsequently, produce solid fuels with better quality [47]. Moreover, it can lower transportation and storage costs [48].

In the last decade, the impact of the torrefaction process upon biomass properties has been detailly studied, and much knowledge has been gained. Reviewing the literature suggests that the torrefaction process of lignocellulosic biomass has been extensively studied, similar to coal-based fuels [49–52], but torrefaction of non-lignocellulosic biomass is rare. Dyjakon and Noszczyk [53] investigated the torrefaction process of horse chestnuts, oak acorns, and spruce cones at conditions mentioned in the above paragraph; solid energy yields were determined on raw and torrefied biomass samples and compared. Zhang et al. [54] torrefied rice straw. Similarly, Chang et al. [48] studied the torrefaction of oil palm solid waste by Fourier transform infrared spectroscopy (FTIR) analysis. Torrefaction efficiency was analyzed during the process of sugarcane bagasse [55,56]. Chen et al. [57] researched the impact of the torrefaction process of sewage sludge that contains the element mercury (Hg) in the sample on the environment. As a result, the 150 wet tons/day torrefaction process of the biomass sample occurred.

*Literature Review*

Previous works indicated that to use torrefied lignocellulosic and non-lignocellulosic biomasses as fuel, the knowledge of physical, chemical, and structural changes caused by the torrefaction process must be studied [58,59]. The properties of torrefied biomass material predominantly rely on the torrefaction severity (TS), which is calculated either by weight loss (WL), severity factor (SF), or torrefaction severity index (TSI). Energy yield, carbon content, and fixed carbon content are mostly shown by weight loss [39]. In addition to weight loss, mass yield and thermal degradation of biomasses are also used as indicators to show TS with an account of thermal degradation depending mostly on the type of feedstock [60]. Thus, to minimize the effect of the nature of the biomass, the researchers defined another index called the torrefaction severity index (TSI), which could be correlated with biomass loss during diverse torrefaction operations [39]. The TSI index under the most severe conditions was also defined [61,62]. It indicated the quality of obtained torrefied biomass material from the lowest to the highest. TSI has been utilized to describe the thermal degradation degree of biomass, EF, energy yield, deoxygenation, dehydrogenation, decarburization, and upgrading energy index [58]. The severity factor (SF) was defined based on temperature and duration of torrefaction [63]. SF was initially conducted to evaluate steam, aqueous, and dilute acid pretreatment [47] but was later introduced to other processes as well. Past studies on the torrefaction process and TSI are summarized in Table 1.

**Table 1.** Past studies on the torrefaction process and TSI.

| Feedstock | Temperature (°C) | Duration (Time, min) | Main Finding | Reference |
|---|---|---|---|---|
| Rubberwood, Gliricidia | 250, 275, 300 | 30, 45, 60 | Calculated EMCI of Rubberwood at 275 °C and 60 min or 300 °C and 45–60 min and Gliricidia at 300 °C and 60 min indicate favourable torrefaction conditions. | [62] |
| Oak waste wood, mixed waste wood, sewage sludge | 220, 240, 260, 280, 300, 320, 340, 400 | 30, 60, 90, 120 | From an energy point of view, the optimal torrefaction temperature is 260 °C, and the optimal torrefaction time is 80 min. When TSI increases, the greater the loss in biomass. | [40] |
| *Eucalypthus grandis* | 210, 230, 250, 270, 290 | 10, 25, 40, 55, 70 | The results were determined by five indexes (weight loss, EF, TS, TSI, and TSF). The obtained results were confirmed to be meaningful for guiding torrefaction operations and reactor design. | [60] |
| Coffee grounds, Chinese medicine residue, algae residue (*Arthrospira plantesis*), and Microalgae residue (*Chlamydomonas* sp. JSC4) | 200, 250, 275, 300 | 15, 30, 45, 60 | Torrefaction severity factor (TSF) can accurately correlate weight loss and torrefaction severity index when optimizing the time exponent. | [14] |
| Coffee grounds, Chinese medicine residue, algae residue (*Arthrospira plantesis*) | 200, 250, 275, 300 | 15, 30, 45, 60 | The results suggested that the quantities of the individual biomass can be predicted via the torrefaction severity index. | [47] |

**Table 1.** *Cont.*

| Feedstock | Temperature (°C) | Duration (Time, min) | Main Finding | Reference |
|---|---|---|---|---|
| Microalgae residue (*Chlamydomonas* sp. JSC4) | 200, 250, 300 | 15, 30, 60 | The results indicate that the torrefaction process has a larger influence on the oxygen and hydrogen losses as a consequence of dehydration and devolatilization. | [64] |
| Almond shell, almonds | 230, 260, 290 | 60, 80, 100 | Condensate mass yields and GCV increased in value for higher torrefaction temperatures and longer times when torrefying raw almond shells into a high-energy, dense fuel source with low moisture contents. | [65] |
| Microalgae residues (*Chlamydomonas* sp. JSC4 and *Chlorella sorokiniana* CY1) | 200, 225, 250, 275, 300 | 40, 60 | The calculated TSI of the two residues are similar to each other; therefore, this parameter may be used to describe the torrefaction extents of various biomass materials. | [63] |

In such context, the torrefaction properties of three different lignocellulosic and non-lignocellulosic biomasses and their mixtures were studied in this work under specific conditions (temperature, time, and atmosphere) with special attention to the physical, chemical, and structural changes (miscanthus, hops, sewage sludge, mixture of sewage sludge and miscanthus, and mixture of sewage sludge and hops). The torrefaction severity index (TSI) was calculated based on temperature and time concerning biomass weight loss. There have already been many studies on the torrefaction behavior of miscanthus and sewage sludge; however, minimal torrefaction studies are reported on the hops, and no torrefaction studies are reported on the mixtures of sewage sludge and miscanthus or hops. Sewage sludge is not directly suitable for usage as a solid fuel, mainly due to low calorific value and relatively poor carbon content. However, its mixing/combining with other raw lignocellulosic materials can significantly improve its fuel properties. The novelty of this study is in the different combinations of several waste materials, such as miscanthus and hops mixtures with sewage sludge, to obtain the optimal torrefaction conditions since many researchers worldwide are facing difficulties in choosing which torrefied biomass material has better quality. Miscanthus and hops, due to their high volatile content, high carbon content, and relatively easy availability, represent a promising type of lignocellulosic biomass for improving the properties of sewage sludge-based biofuels. Therefore, the obtained data can provide essential information for the future utilization of second-generation biofuels and can be useful for understanding the basic phenomena related to the torrefaction process and sewage sludge disposal.

## 2. Experimental Part

### 2.1. Biomass Feedstock and Sample Preparation

In the present study, two lignocellulosic biomass samples and one non-lignocellulosic biomass served as the samples for the torrefaction experiment: miscanthus (100%), hops (100%), and sewage sludge (100%), which originated from Slovenia. Miscanthus (M) (*Miscanthus* × *giganteus*) was collected in the Podravje region and was received in the following dimensions: 1.5 cm × 0.3 cm × 0.5 cm. Hops (H) (*Humulus lupulus*) was collected in the region of Savinjska and was received directly after harvesting, with the white rope included (type TP 1000, UVS 1200). Both hops and miscanthus were chosen in this study as they present Slovenian biomass diversity and potential resources for use in

the thermochemical processes. Sewage sludge (SS) was received from a publicly owned wastewater treatment plant based in the region of Podravje. As stated in the work of Jayaraman et al. [31], the composition of dried sewage sludge is almost comparable with miscanthus; therefore, dried sewage sludge was used in this study. SS was also used in this study because Slovenia has faced severe environmental and cost problems regarding the disposal of SS in the last few years [66]. In total, 8,000,000 tons of municipal SS is produced annually in the country, with costs of more than EUR 127/ton. So far, 90% of the produced SS has been taken to Hungary for processing, but lately, researchers across the country are attempting to discover alternative ways for its utilization and re-usage [67]. Additionally, the following mixtures were used in this work: a mixture of sewage sludge and miscanthus (50:50%) (SS + M) and a mixture of sewage sludge and hops (50:50%) (SS + H). As stated, to date, no research on the torrefaction of mixtures of sewage sludge and miscanthus or hops has been conducted.

The biomass was dried and prepared according to standard protocols [67,68], and the homogenized material is presented in Table 2.

**Table 2.** Physical appearances of raw and torrefied biomass materials.

| Biomass | (M) Miscanthus (100%) | (H) Hops (100%) | (SS) Sewage Sludge (100%) |
|---|---|---|---|
| Photos | 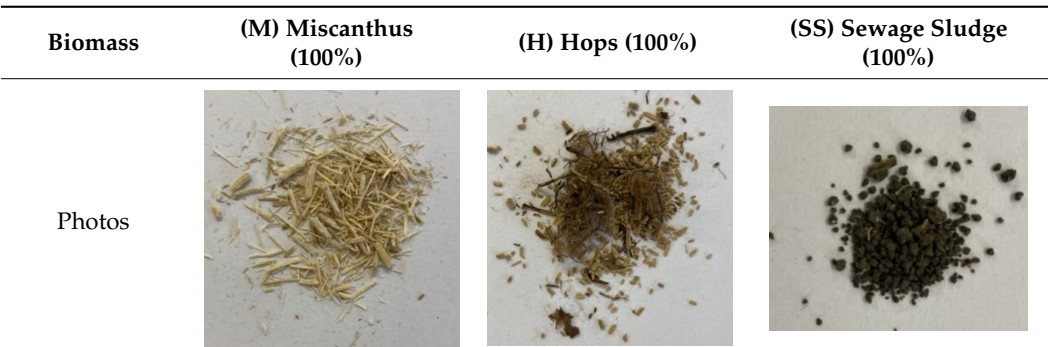 | | |

### 2.2. Torrefaction Process of the Raw Biomass

The torrefaction process was conducted directly in the thermal analyzer equipment Mettler Toledo TGA/DSC 3⁺ STAR System (Greifensee, Switzerland). Thermal conversion was conducted under a nitrogen ($N_2$) atmosphere (25 °C, 20 mL/min). All experiments were carried out in an alumina crucible, and up to 40 mg of sample were used in the experiment, respectively. The thermal decomposition of the samples was performed according to the protocol described in the work of Zhang et al. [11]: firstly, the samples were heated from room temperature, 30 °C, to 105 °C with a heating rate of 25 °C/min, respectively. At a temperature of 105 °C, each sample was held for 10 min to remove the moisture, and then the samples were heated to the chosen torrefaction temperature, which was 250, 300, and 350 °C. After that, the specific torrefaction temperature was maintained for 10, 30, and 60 min. The course of the experimental work is presented in Figure 1, and the torrefaction setup is shown in Figure 2.

To determine the torrefaction performance, for each sample, mass (MY) and energy yields (EY) were calculated together with enhancement factor (EF) and the energy–mass co-benefit index (EMCI) using Equations (1)–(4).

$$\text{MY (\%)} = \frac{\text{mass}_{\text{torrefied sample}}}{\text{mass}_{\text{raw sample}}} \cdot 10, \tag{1}$$

$$\text{EF} = \frac{\text{HHV}_{\text{torrefied sample}}}{\text{HHV}_{\text{raw sample}}}, \tag{2}$$

$$\text{EY (\%)} = \left( \text{MY} \cdot \frac{\text{HHV}_{\text{torrefied sample}}}{\text{HHV}_{\text{raw sample}}} \right) = \text{MY} \cdot \text{EF} \tag{3}$$

$$\text{EMCI} = \text{EY} - \text{MY}. \tag{4}$$

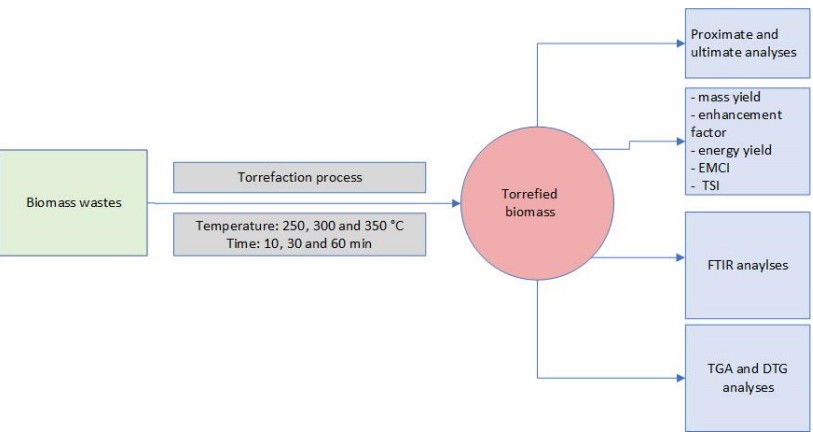

**Figure 1.** Schematic diagram of the experiment.

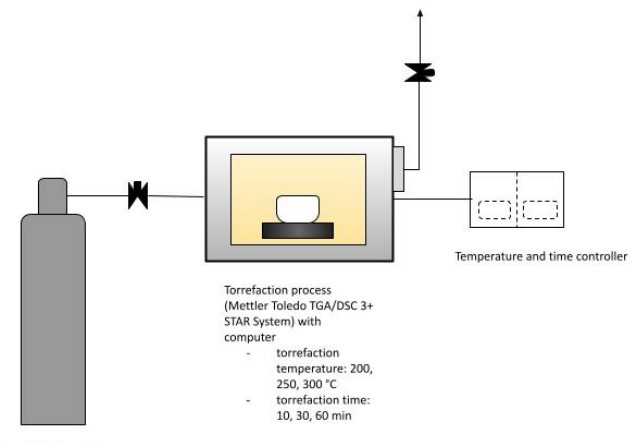

**Figure 2.** Torrefaction setup.

### 2.3. Analytical Methods

The basic properties of biomasses, such as proximate and ultimate analyses and calorific values, were determined. The proximate and ultimate analyses of dry samples were performed as described in our previous papers [52,69] and are represented by the equations below (Equations (5)–(8)) [7].

$$\text{MC (wt.\%)} = \left(\frac{A - B}{A - C}\right) \cdot 100, \tag{5}$$

$$\text{AC (wt.\%)} = \left(\frac{B - C}{A - C}\right) \cdot 100, \tag{6}$$

$$\text{VM (wt.\%)} = \frac{(A - B) - (MC \cdot (A - C))}{(A - C) \cdot (100 - MC)} \cdot 100, \tag{7}$$

$$\text{FC (wt.\%)} = 100 - MC - AC - VM, \tag{8}$$

where AC is ash, TC is fixed carbon, Vm is volatile matter, MC is moisture content, A is the sum of the crucible mass and sample mass, B is the sum of the crucible mass and sample mass after drying, and C is the sum of the empty crucible mass.

Standard procedure was applied for the determination of quantities in Equations (5)–(7) following refs. [70–72]. The weight percentage of O was calculated using Equation (9):

$$\text{O (wt.\%)} = 100 - \text{C (wt.\%)} - \text{H (wt.\%)} - \text{N (wt.\%)} - \text{S (wt.\%)} - \text{AC (wt.\%)}. \tag{9}$$

Higher heating values (HHV) were detected by a bomb calorimeter (IKA C6000 Isoperibol (Staufen, Germany) following the UNI EN 14918. To ensure the measurement quality, all analyzers were calibrated at regular intervals. For each analysis, up to 20 mg of each biomass material was used. The method was carried out in triplicate in order to calculate the standard deviation.

To investigate the thermal degradation characteristics of biomasses, thermogravimetric (TGA) and derivative thermogravimetric (DTG) analyses of the samples were conducted on raw and torrefied biomass samples using the thermogravimetric analysis described above in Section 2.2. The samples were heated at the constant heating rate, 25 °C/min under a nitrogen medium (25 °C, 20 mL/min), from 30 °C until they reached the final torrefaction temperature. With each biomass sample, three identical experiments were performed to verify repeatability, and the results show that the absolute differences between the three identical experiments are less than 8%. Lastly, the chemical functionalities of the biomass materials (i.e., the raw and torrefied samples) were characterized by a Fourier transform infrared (FTIR) spectrometer, Thermo Scientific, Nicolet iS50 FTIR (Waltham, MA, USA), with an attenuated total reflectance (ATR) sampling technique. The spectra were recorded between 400 and 4000 cm$^{-1}$ by averaging 16 scans with a spectral resolution of 4 cm$^{-1}$. The most severe torrefaction conditions (350 °C, 60 min) were selected for comparison purposes in both TGA and FTIR analyses.

The torrefaction severity index (TSI) introduced in this work is defined as shown in Equation (10):

$$\text{TSI} = \frac{\text{WL}_{T,t}}{\text{WL}_{350\,°C,\,60\,\text{min}}} = \frac{100 - \text{MY}_{T,t}}{100 - \text{MY}_{350\,°C,\,60\,\text{min}}}, \tag{10}$$

where WL represents the weight loss of the torrefied biomass at the specific torrefaction temperature (T) and time (t), and MY represents the mass yield.

## 3. Results

### 3.1. Properties of Raw Biomass Samples

The biomasses were characterized based on the moisture content, fixed carbon content, volatile matter content, ash content, higher heating value (HHV), and elemental analysis. Table 3 summarizes the values obtained during the experimental analysis performed for the characterization of the different biomasses. The obtained results from the proximate analyses are as follows: volatile matter content for the miscanthus, hops, sewage sludge, and mixtures were $82.8 \pm 3.3$, $82.2 \pm 0.3$, $56.0 \pm 2.2$, $67.9 \pm 2.7$, and $62.1 \pm 2.5$ wt.%, respectively. The results of volatile compounds were higher in the lignocellulosic biomasses than in non-lignocellulosic biomass, which indicated that lignocellulosic biomasses possessed higher reactivity compared to non-lignocellulosic materials [73,74]. Consequently, the contents of fixed carbon were significantly lower (<12 wt.%) than those of the volatile matter, especially in the lignocellulosic biomasses. The ash contents were in the range of $2.8 \pm 0.2$, $3.2 \pm 0.2$, $16.3 \pm 1.0$, $11.7 \pm 0.7$, and $13.5.0 \pm 0.8$ wt.% for each biomass, respectively. The ash contents of sewage sludge and mixtures of sewage sludge and miscanthus/hops were largely higher than others. It is believed that this was due to the higher amount of ash present in the original raw sewage sludge biomass sample. Moisture contents varied from $9.2 \pm 0.3$ to $12.4 \pm 0.4$ wt.% for each biomass separately. HHV was the highest in the miscanthus sample ($18.9 \pm 1.0$ MJ/kg), whereas the lowest was in the pure SS biomass sample ($14.98 \pm 0.7$ MJ/kg). The contents of carbon in the miscanthus, hops, SS, and mixtures were $45.1 \pm 1.4$, $42.1 \pm 1.3$, $34.7 \pm 1.0$, $40.4 \pm 1.2$, $38.9 \pm 1.1$ wt.%, respectively. These results came from the basic structure of the biomasses. Both miscanthus and hops are mainly composed of cellulose, hemicelluloses, and lignin, whilst SS contain more inorganic elements, such as N, P, K, Ca, and Mg, and other heavy metals and other materials that are present in the industrial wastes [75]. It has been stated in many works [76,77] that the torrefaction process leads to the enrichment of the carbon content whilst the content of hydrogen decreases. The contents of hydrogen were low, as confirmed in the literature [78] and were in the range from $3.7 \pm 0.1$ to $5.2 \pm 0.1$ wt.%. The greatest hydrogen content was

found in the pure SS, and the lowest was found in the raw miscanthus sample. The nitrogen contents varied from $0.8 \pm 0.02$ to $4.8 \pm 0.1$ wt.% and oxygen contents from $36.3 \pm 1.0$ to $47.4 \pm 1.45$ wt.%. The contents of sulphur were unusually very low ($0.05 \pm 0.01$, $0.03 \pm 0.01$, $0.82 \pm 0.04$, $0.31 \pm 0.01$, $0.47 \pm 0.01$ wt.%) in accordance with literature [42,79]. These differences may later affect the torrefaction process obtained from biomass materials and the characteristics themselves. All results agree with other works [33,41,80–83].

**Table 3.** Physicochemical properties of raw biomass samples.

| Analysis | | M | H | SS | M + SS | H + SS |
|---|---|---|---|---|---|---|
| Proximate analysis (wt.%, dry basis) | Fixed carbon | $3.89 \pm 0.08$ | $2.01 \pm 0.04$ | $15.08 \pm 0.30$ | $7.41 \pm 0.15$ | $7.14 \pm 0.14$ |
| | Volatile matter | $82.79 \pm 3.31$ | $82.23 \pm 0.33$ | $56.01 \pm 2.24$ | $67.89 \pm 2.72$ | $62.13 \pm 2.49$ |
| | Ash content | $2.83 \pm 0.17$ | $3.18 \pm 0.19$ | $16.33 \pm 0.98$ | $11.73 \pm 0.70$ | $13.48 \pm 0.80$ |
| Moisture content (wt.%, dry basis) | | $9.21 \pm 0.27$ | $11.01 \pm 0.33$ | $12.35 \pm 0.37$ | $10.34 \pm 0.31$ | $11.42 \pm 0.34$ |
| Elemental analysis (wt.%, dry basis) | C | $45.11 \pm 1.35$ | $42.12 \pm 1.27$ | $34.67 \pm 1.04$ | $40.39 \pm 1.21$ | $38.98 \pm 1.17$ |
| | H | $3.71 \pm 0.11$ | $4.54 \pm 0.14$ | $5.19 \pm 0.15$ | $4.67 \pm 0.14$ | $5.08 \pm 0.15$ |
| | N | $0.80 \pm 0.02$ | $3.49 \pm 0.10$ | $4.79 \pm 0.14$ | $3.94 \pm 0.12$ | $3.72 \pm 0.11$ |
| | O | $46.13 \pm 1.38$ | $47.43 \pm 1.42$ | $38.20 \pm 1.15$ | $36.33 \pm 1.09$ | $38.27 \pm 1.15$ |
| | S | $0.05 \pm 0.01$ | $0.03 \pm 0.01$ | $0.82 \pm 0.04$ | $0.31 \pm 0.01$ | $0.47 \pm 0.01$ |
| Energy content (wt.%, dry basis) | HHV (MJ/kg) | $18.91 \pm 0.95$ | $16.56 \pm 0.82$ | $15.21 \pm 0.76$ | $17.04 \pm 0.85$ | $15.48 \pm 0.77$ |

### 3.2. Torrefaction Performance and Severity Index

The results of HHV and the calculated mass yields, enhancement factors, energy yields, and EMCI of the five torrefied biomass samples in response to different torrefaction temperatures (i.e., 250, 300, and 350 °C) and different torrefaction times (i.e., 10, 30, 60 min) are illustrated in Figure 3 and Table 4.

**Table 4.** HHV (MJ/kg) and EMCI of studied biomasses with respect to temperature and time.

| Material | T (°C) | HHV (MJ/kg) | | | EF | | | EMCI | | |
|---|---|---|---|---|---|---|---|---|---|---|
| | | 10 min | 30 min | 60 min | 10 min | 30 min | 60 min | 10 min | 30 min | 60 min |
| M | 250 | 19.59 | 19.80 | 20.26 | 1.04 | 1.05 | 1.07 | 3.05 | 3.73 | 5.38 |
| | 300 | 19.34 | 19.94 | 20.3 | 1.02 | 1.05 | 1.07 | 1.47 | 2.83 | 2.97 |
| | 350 | 19.00 | 19.7 | 20.32 | 1.00 | 1.04 | 1.07 | 0.163 | 1.37 | 2.39 |
| H | 250 | 16.98 | 17.12 | 17.38 | 1.03 | 1.03 | 1.05 | 1.84 | 2.31 | 3.24 |
| | 300 | 16.71 | 18.78 | 20.9 | 1.01 | 1.13 | 1.26 | 0.45 | 6.06 | 10.45 |
| | 350 | 17.33 | 17.65 | 19.18 | 1.05 | 1.07 | 1.16 | 2.25 | 3.08 | 8.45 |
| SS | 250 | 15.96 | 15.97 | 16.11 | 1.05 | 1.05 | 1.06 | 4.17 | 4.07 | 4.67 |
| | 300 | 15.77 | 17.33 | 18.86 | 1.04 | 1.14 | 1.24 | 2.41 | 8.67 | 14.46 |
| | 350 | 15.46 | 16.11 | 16.79 | 1.02 | 1.06 | 1.10 | 0.97 | 3.39 | 5.85 |
| M + SS | 250 | 17.91 | 17.81 | 18.01 | 1.05 | 1.05 | 1.06 | 4.33 | 3.68 | 4.43 |
| | 300 | 17.43 | 17.92 | 18.53 | 1.02 | 1.05 | 1.09 | 1.55 | 3.09 | 4.71 |
| | 350 | 17.29 | 17.12 | 18.32 | 1.01 | 1.05 | 1.08 | 0.69 | 0.21 | 3.35 |
| H + SS | 250 | 15.78 | 15.42 | 15.96 | 1.02 | 1.00 | 1.03 | 1.62 | 3.28 | 2.42 |
| | 300 | 16.36 | 16.87 | 17.90 | 1.06 | 1.09 | 1.16 | 3.66 | 5.48 | 9.28 |
| | 350 | 15.29 | 15.10 | 15.36 | 1.00 | 1.00 | 1.00 | 0.66 | 1.28 | 0.39 |

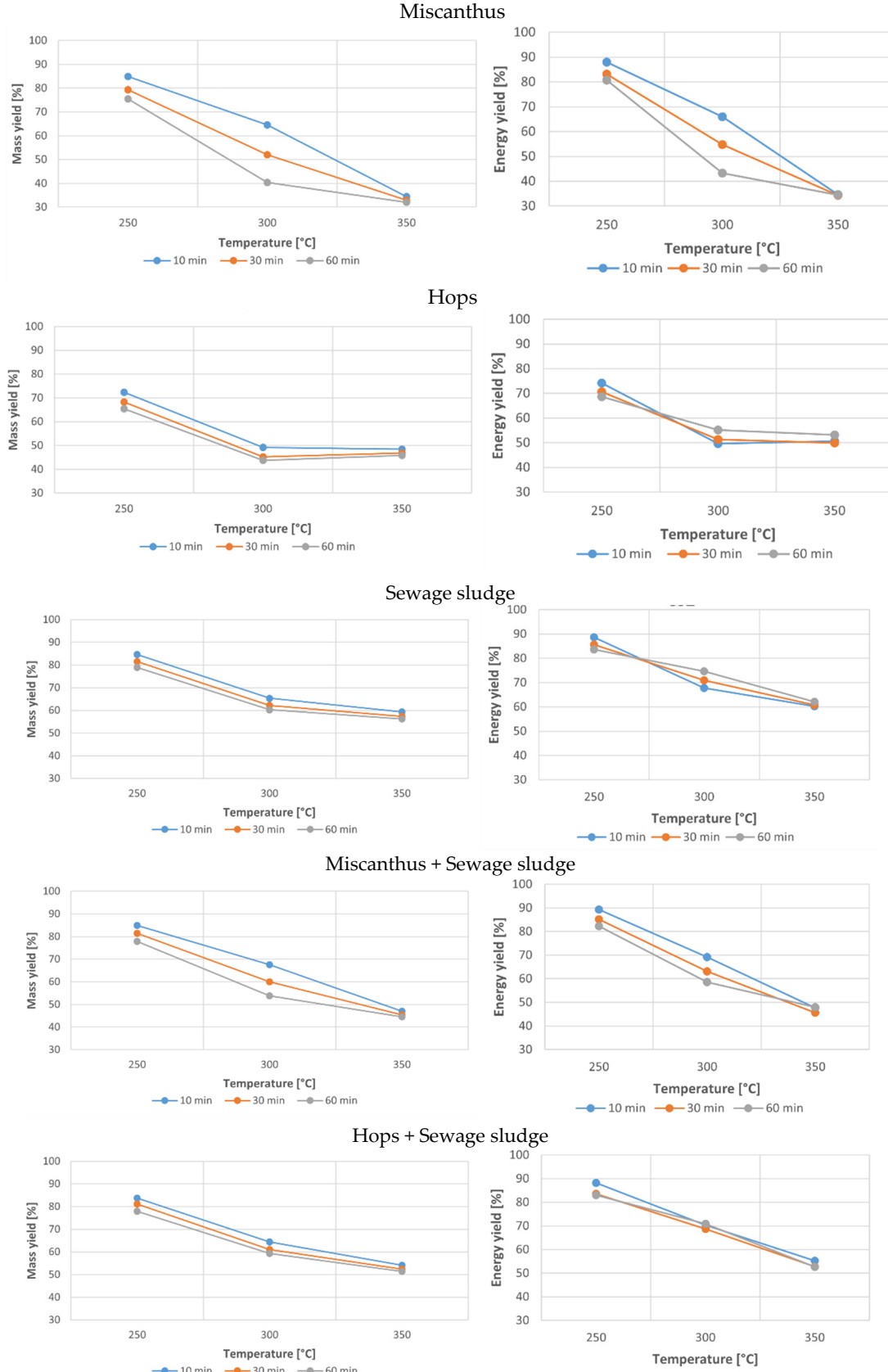

**Figure 3.** Mass yield and energy yield with respect to torrefaction temperature and time for all five materials.

Figure 3 shows the relationship between mass and energy yields for all studied torrefaction temperatures and times, respectively. In this work, the most severe torrefaction conditions were defined to be 350 °C and 60 min. The plots in Figure 3 show that mass yield decreased in all five torrefied biomass samples as torrefaction temperature increased, respectively, which is completely in agreement with other works in the literature regardless of the type of studied biomass materials [59,84,85]. Additionally, when torrefaction time increased, the mass yields decreased. Both torrefaction temperature and time have a big impact on the biomasses during the torrefaction process, with torrefaction temperature having a more significant one [86]. When the torrefaction temperature increased, moisture was removed from the samples, and mostly all volatiles were decomposed. Only a minor impact was observed on the biomasses when torrefaction time was increased from 10 to 30 min and then to 60 min. When increasing the torrefaction time from 10 min to 30 min and later to 60 min, the most significant impact was seen during the torrefaction of miscanthus and hops biomass samples; during the torrefaction of other biomass samples, only a minor impact was observed. For torrefaction time 10 min, the calculated mass yields were 84.92–34.41%, 72.36–48.38%, 84.4–59.28%, 83.72–54.15%, and 84.90–46.97% at 250, 300, and 350 °C for miscanthus, hops, SS, and mixtures, respectively. For torrefaction time 30 min, those values were 79.37–32.91%, 68.28–46.81%, 81.49–57.36%, 81.25–52.38%, and 81.47–45.42%, and for torrefaction time 60 min they were approximately 75.41–32.10%, 65.46–44.64%, 78.93–56.29%, 77.97–51.37%, and 77.89–44.54%. The lowest mass yields were obtained at the most severe torrefaction conditions. The same was observed in the work of Zhang et al. [47] and Nepal [19], whereas from their results, Simonič et al. [40] could also predict the optimal torrefaction temperature and time: approximately 260 °C and 30 min. The calculated mass yields were similar between lignocellulosic biomass samples; however, they were a little different from the calculated SS mass yields, which were generally slightly lower. At the beginning of the process, mass yields in the SS biomass sample decreased more notably than later during the process. This was also seen in the work of Poudel et al. [87], who investigated the torrefaction of SS at temperatures ranging between 150–400 °C and torrefaction time varying from 0 to 50 min. In their work, the obtained mass yields decreased when the torrefaction temperature was increased, as well as when the torrefaction time was raised. At the beginning of torrefaction, a significant mass loss was observed, while the change of mass yield was not significant with a longer torrefaction time. When adding lignocellulosic biomass to SS in our work, the mixtures then observed similar mass yield as lignocellulosic biomass, which potentially means that the properties of the fuel were improved. During the process, the HHV values were also determined. HHV values increased with both torrefaction temperature and time (Table 4). As already stated in this work, during the torrefaction process, the HHV was increased as a result of the decrease in oxygen content and increase in carbon and fixed carbon contents. The enhancement factor (EF) reflects the change in HHV during the torrefaction process. The results of EF calculated in this work are listed in Table 4. The results showed that EF increased as torrefaction temperature and time increased, which means that HHV values improved during the process. Furthermore, energy yield was proportional with mass yield and HHV values; therefore, energy yield was determined next. Similar mass yield was observed in the calculated energy yields, which decreased with torrefaction temperature and time (Figure 3). For torrefaction time 10 min, the calculated energy yields were 87.96–34.57%, 74.19–50.62%, 88.71–60.25%, 85.34–53.48%, and 89.23–47.66%, respectively, for each biomass, for torrefaction time 30 min, the calculated energy yields were 83.1–34.28%, 70.59–49.89%, 85.56–60.75%, 80.83–51.09%, and 85.15–45.63%, and for torrefaction time 60 min those values were varying between 80.79–34.49%, 68.70–53.10%, 83.60–62.13%, 80.38–50.97%, and 82.32–47.88%, respectively. Again, the most significant decrease was seen during the torrefaction of miscanthus and hops biomass samples, whereas during the torrefaction of other biomass samples minor decreases were observed. The decrease in energy yield was smaller than the decrease in mass yield, because the increased torrefaction temperature helped to improve the energy density of torrefied biomass samples.

In this work, TSI showed the degree of weight loss of the biomass at the different torrefaction conditions. The profiles of TSI and torrefaction temperature are shown in Figure 4 for specific torrefaction times. TSI was determined at the highest torrefaction temperature studied in this work (350 °C) and the longest torrefaction time (60 min). Based on the literature and as already mentioned, these values were in the range of 0–1 [19], where value 0 presented that the torrefaction process had not yet begun, and value 1 meant that the torrefaction process reached the highest torrefaction temperature and time. The results of this work are in agreement with other works [40,47].

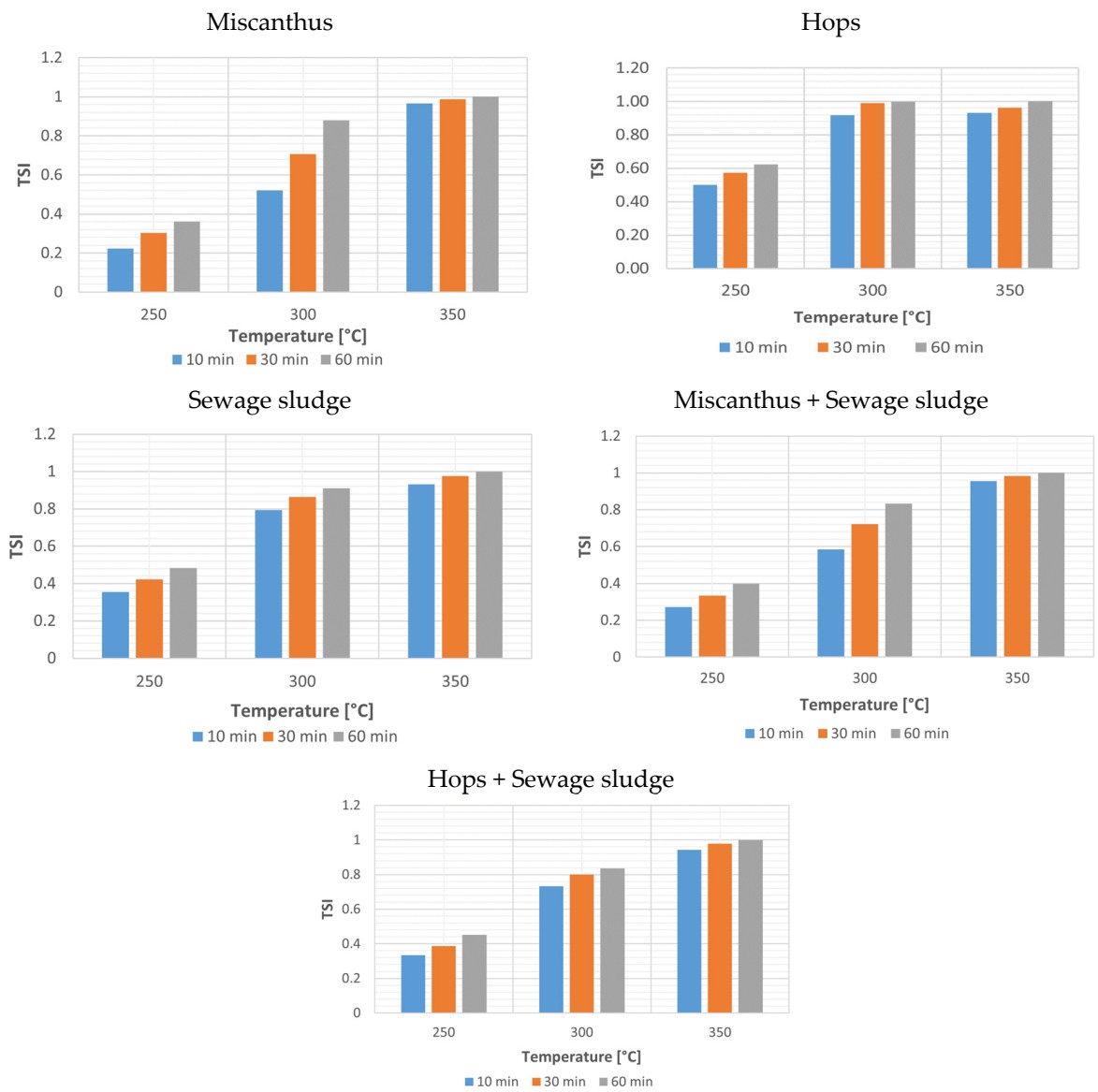

**Figure 4.** TSI with respect to torrefaction temperature and time for all five materials.

In addition, the same was stated for the correlation between TSI and energy yield, which is presented in Figure 5.

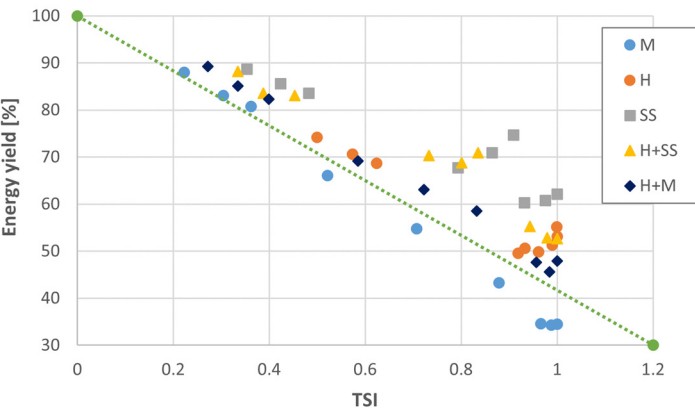

**Figure 5.** TSI and energy yield with respect to torrefaction temperature and time.

Additionally, the energy–mass co-benefit index (EMCI) was calculated. The results for each biomass sample are presented in Table 4. EMCI is an index that could describe the economic aspect of the torrefaction process, and its definition says that it represents the difference between the energy yield and the mass yield at the lowest possible torrefaction severity. The higher the EMCI, the higher the energy density, and the lower the volume of the torrefied biomass [62]. Our results show that the highest EMCI was calculated for SS and the mixture of SS and miscanthus, 14.46 and 11.57 at 300 °C, respectively. Under other torrefaction conditions, the calculated EMCI indexes were slightly lower. From the obtained results, sewage sludge stands out the most. It is believed this is due to the lack of lignocellulosic components in the sample.

### 3.3. TGA and DTG Analysis

Thermogravimetric analyses for all materials at severity torrefaction conditions (350 °C and 60 min) are presented in Figure 6. The measurements were performed in an inert atmosphere (20 mL/min of $N_2$) in the range of 30–350 °C at a heating rate of 25 °C/min. In the beginning, the material was heated to 105 °C and left for 10 min at that temperature, then the temperature was raised to 350 °C and stayed at that value for 60 min. The decomposition of all samples occurred in two stages. The first stage was due to the evaporation of moisture and lasted up to approximately 200 °C. In this stage, there were no differences in mass loss between the individual materials. In the second stage, which took place between 200 °C and 350 °C, the combustion of lighter and heavier fractions of volatile substances occurred. In the lignocellulosic biomass samples, the degradation of hemicelluloses, cellulose, and lignin occurred, whereas the sewage sludge sample did not contain these main compounds. Firstly, hemicellulose and cellulose decomposed, followed by lignin. Chen et al. [51] found that each raw lignocellulosic biomass sample contained between 32–45% cellulose, 19–25% hemicellulose, and 14–26% lignin, and that these components started to degrade between 200–315 °C, 315–400 °C, and 250–500 °C, respectively, which can also be seen in the plots below. The results of this work are in agreement with our previous works and some works reported in the literature [4,69,88]. In these works, it was also found and confirmed that according to the obtained results of all analyses, the optimal temperature for the torrefaction process is approximately 250 °C, where the greatest increase in HHV was obtained. The highest mass loss was observed 10 min after the torrefaction process took place; therefore, it could be stated that the optimal time for the studied biomasses is 10 min, depending on the biomass sample sizes.

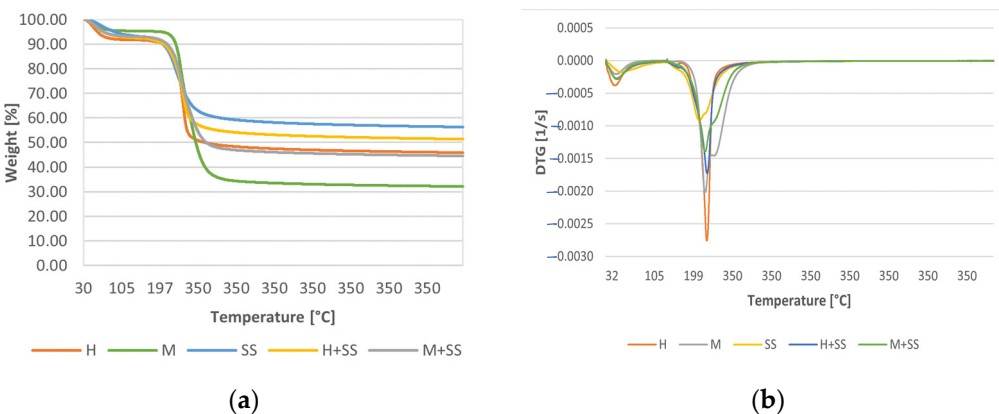

**Figure 6.** TGA (**a**) and DTG (**b**) curves for all materials, torrefied at 350 °C for 60 min.

*3.4. FTIR Analysis*

FTIR spectra of raw samples (miscanthus, hops, SS, and their mixtures) and those torrefied at the most severe torrefaction conditions (350 °C, 60 min) are shown in Figure 7. Significant differences in chemical structure, i.e., functional groups, can be observed from the spectra of thermally treated biomass samples due to changes in the chemical structure of the lignin, hemicellulose, and cellulose [19]. The raw samples showed peaks, typical for lignocellulosic biomass: the area between 3600 and 3200 cm$^{-1}$ is associated with the O–H bonds of cellulose, peaks between 2900 and 2800 cm$^{-1}$ represent the aliphatic C–H bonds of cellulose and hemicellulose, peaks in the area of 1700–1500 cm$^{-1}$ are attributed to the stretching vibrations of the C=O group in carboxylic acids in hemicelluloses and C=C aromatic rings of lignin, and peaks between 1500 and 1100 cm$^{-1}$ indicate C–C, C–O, and C–H bonds. The broad band at ~1030 cm$^{-1}$ was observed in the case of hops and miscanthus samples attributed to the stretching vibration of C–O and C–O–C bonds of the lignin, while in the case of SS, it can additionally represent P–O bond. Peaks associated with the amide group of proteins and nitrogen compounds of SS occurred in the area between 1550 and 1400 cm$^{-1}$. The FTIR spectra of raw materials, such as SS [4] and miscanthus [89], agree with the spectra of the same or similar materials presented in other studies.

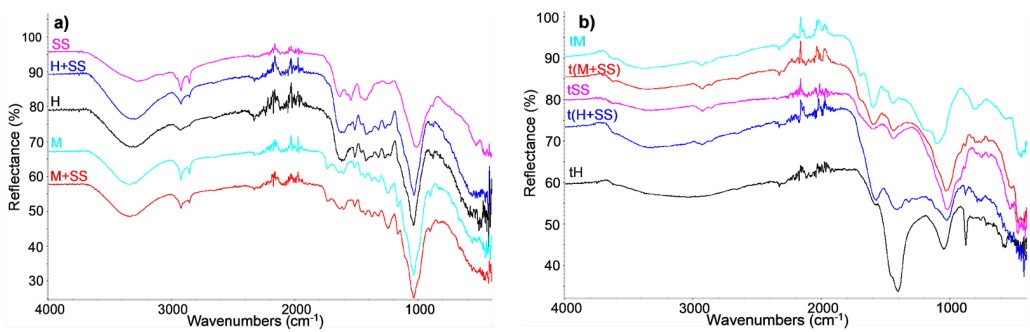

**Figure 7.** FTIR spectra of (**a**) raw samples and (**b**) torrefied samples (350 °C, 60 min).

After torrefaction, the intensity of peaks reflecting the O–H and C–H bonds decreased due to the degradation of cellulose and hemicellulose because of dehydroxylation and condensation reactions. The peaks of C=C and C=O groups representing lignin also showed a noticeable change, indicating that decarboxylation and acetylation reactions occurred in lignin compounds during torrefaction [90]. Torrefied hops, in comparison to other torrefied samples, showed a sharp peak at 870 cm$^{-1}$, most likely indicating aromatic C-H or hetero-aromatic compounds. According to FTIR results, it can be concluded that the torrefaction promotes the decomposition of mainly hydrogen and oxygen-containing functional groups. Similar changes in the chemical structure of torrefied biomass were also reported for other

types of biomasses, such as rubber and Gliricidia wood [62], spent coffee grounds [19], corncobs [90], and others.

## 4. Conclusions

In this study, the torrefaction process under a nitrogen atmosphere was investigated on five different biomasses, lignocellulosic and non-lignocellulosic, in temperatures ranging between 250–300 °C and time within the range of 10–60 min. Several analytical methods were used to determine the optimum torrefaction temperature and time for miscanthus, hops, sewage sludge, and mixtures of sewage sludge and miscanthus or hops. The results showed that as the torrefaction temperature increased, the mass and energy yield decreased while HHVs increased. The same was observed when the torrefaction time increased for all studied biomasses. Torrefaction severity index (TSI) showed that the higher the losses in biomass samples, the greater the value of TSI. Additionally, the effect of the torrefaction process was demonstrated by thermal degradation through TGA and DTG analysis and by structural changes of the biomasses through FTIR. The analysis of the FTIR spectra of torrefied biomass samples confirmed the chemical changes of the hemicellulose, cellulose, and lignin components due to thermal degradation. According to the results, it can be confirmed that the torrefaction process improves the properties of raw miscanthus and hops, while the torrefaction properties are negligible for the pure SS biomass sample.

In our future research, the kinetics of pyrolysis of the same materials will be evaluated.

**Author Contributions:** Conceptualization, methodology, software, validation, investigation, formal analysis, writing—original draft: M.I.; methodology, validation, formal analysis, data curation, writing—review and editing, supervision: D.U.; software, validation, investigation, writing—original draft A.P.; software, validation, investigation, writing—original draft: J.S.; supervision: D.G.; supervision: M.S. All authors have read and agreed to the published version of the manuscript.

**Funding:** This research received no external funding.

**Institutional Review Board Statement:** Not applicable.

**Informed Consent Statement:** Not applicable.

**Data Availability Statement:** Not applicable.

**Conflicts of Interest:** The authors declare no conflict of interest.

## Abbreviations

| | |
|---|---|
| AC | Ash content |
| DTG | Derivative thermogravimetric analysis |
| EF | Enhancement factor |
| EMCI | Energy–mass co-benefit index |
| EY | Energy yield |
| FC | Fixed carbon |
| FTIR | Fourier transform infrared spectroscopy |
| H | Hops |
| HHV | Higher heating value (MJ/kg) |
| M | Miscanthus |
| MY | Mass yield |
| RES | Renewable energy sources |
| SF | Severity factor |
| SS | Sewage sludge |
| VM | Volatile matter |
| TGA | Thermogravimetric analysis |
| T | Temperature |

| t | time |
| TS | Torrefaction severity |
| TSI | Torrefaction severity index |
| WL | Weight loss |

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
