# Peer review of "Improving Lignocellulosic and Non-Lignocellulosic Biomass Characteristics through Torrefaction Process"

_applsci, doi:10.3390/app122312210_

Round 1

Reviewer 1 Report

Interesting article on improving biomass characteristics though torrefaction process. In the era of changes taking place in the world, research on the improvement of biomass quality takes on a new meaning.

The article describes five types of biomass: miscanthus, hops, sewage sludge, and mixtures of sewage sludge and miscanthus or hops, which are available locally.

It has been shown and confirmed that that the torrefaction process improves the properties of raw miscanthus and hops, while the torrefaction properties are negligible for the pure SS biomass sample. The results  shows that adding a lignocellulosic biomass to the samples is recommended.

The literature review is very rich, it indicates the most important directions for the development of work in the subject. More than 90 analysed articles provide an nice overview and allow reader to understand problem. The work planned for success enables the achieving good results.  The included tables and figures are essential and are easy to read.

The text is written very well, I did not find any spelling or grammatical problems. Although I am not able to guarantee that everything is ok, I only found in line 185 “it which originated from the RS” I think it should be RES. As in the 5. Abbreviations I could only find RES- Renewable energy sources

Author Response

Thank You for Your kind review.

In line 185 should be Slovenia. It was corrected.

Reviewer 2 Report

The paper’s premise was to analyze the impact of torrefaction on miscanthus, hops and sewage sludge and the mixtures of these biomasses.

The paper was denied, due to an inherently flawed scientific approach, lack of novelty and insufficient as well as inadequate interpretation of the results.

The authors claim that the novelty of the paper is the analysis of the biomass mixtures. However, the reason why the high-value biomasses and the sewage sludges were mixed in the first place was not stated. The second novelty was to determine the optimal temperature and residence time of the torrefaction process. The authors claim that these are reached at 250°C and 10min, respectively. However, for the optimal temperature of 250°C not one explanation was given. The TGA analysis was the only result which was given for the verification of an optimal residence time. The last claimed novelty was the TSI value. Here the results showed “that TSI could correlate easily with mass yield”. This is stated multiple times in the paper. As the mass yield is the only parameter in the TSI, this statement seems absolutely trivial. For these given reasons, the paper lacks any type of novelty.

The second reason for denying the paper, was the apparent faults in the methodology seen in table 3. In the table, the properties of the mixtures have such a high deviation from the single biomasses that the methodology has to be incorrect and unscientific. In addition to that, there are no standard deviations given throughout the results and interpretations.

Due to these given reasons, the paper should not be published and lacks novelty or scientific integrity for revision.

Author Response

Thank You for the detailed review. Regarding the novelty: the paragraph (please see lines169-181) has been improved to emphasize the novelty of the research. The answer why the high-value biomass and the sewage sludge were mixed is the following: sewage sludge is not directly suitable for solid fuel, due to low calorific value and relatively poor carbon content, while combining it with other raw ligno-cellulosic material can significantly improve the fuel properties. Miscanthus and hops represent a promising type of ligno-cellulosic biomass for the improving the properties of sewage sludge-based biofuels, due to high volatiles as well as carbon content and they are readily available.

The reviewer doubted about the optimal temperature of 250 °C as not one explanation was given. The optimal temperature of 260 °C was determined for raw materials without sludge as stated in the “Abstract”. The discussion which emphasizes the optimum temperature and time was strengthened (please see lines 455-462). Optimal conditions were determined based on results on HHV, proximative analysis in connection with C content, TGA analysis. The work also confirmed that the torrefaction process improves the properties of raw Miscanthus and hops samples, while the torrefaction properties are negligible for the pure SS biomass sample. Thus, when miscanthus or hops were combined with the sewage sludge, the properties of torrefied mixture were improved in comparison with that of torrefied SS.

It is further stated in lines 517-518: “the optimum conditions may vary slightly depending on the initial condition of the raw materials, the amount of sample used, the heating rate, etc.” Therefore, the exact temperature is very difficult to determine due to big influence of the initial conditions.

The statement “that TSI could easily correlate with mass yield” was just a cue to determine correlations. Since the statement might seem trivial, it was withdrawn.

Regarding the apparent faults in methodology seen in Table 3, authors came to the same conclusion as reviewer by careful observations of the data. The measurements were carried out by an external laboratory, and apparently there were some random errors during the measurements and therefore, the measurements were re-run in three replicates. Additionally, standard deviations were added as suggested by reviewer. Indeed, the results are now realistic. Please see Table 3 and text colored yellow in Sections 3.1 and 3.2.

All issues have been carefully considered, and the authors hope that the manuscript now shows scientific integrity.

Round 2

Reviewer 2 Report

The revisions made by the authors were accepted by the reviewer, as the biggest issue were the errors in the analysis table and missing justifications of the temperature optimum.